# Zero-Shot Text-to-Speech from Continuous Text Streams

## Abstract

Existing zero-shot text-to-speech (TTS) systems are typically designed to process complete sentences and are constrained by the maximum duration for which they have been trained. However, in many streaming applications, texts arrive continuously in short chunks, necessitating instant responses from the system. We identify the essential capabilities required for chunk-level streaming and introduce L3Speech, a stream-aware model that supports non-stop generation, text-audio stream synchronization, and seamless transitions between short speech chunks. To achieve these, we propose (1) adopting Mamba, a class of sequence modeling distinguished by linear-time decoding, which is augmented by cross-attention mechanisms for conditioning, (2) utilizing rotary positional embeddings in the computation of cross-attention, enabling the model to process a non-stop text stream by sliding a window, and (3) decoding with semantic guidance, a technique that aligns speech with the transcript during inference with minimal overhead. Experimental results demonstrate that our models are competitive with state-of-the-art language-model-based zero-shot TTS models, while also providing flexibility to support a wide range of streaming scenarios.

## 1 Introduction

In recent years, significant advancements have been made in the field of text-to-speech (TTS), evidenced by reports of human parity across both single-speaker (Tan et al., 2024) and zero-shot scenarios (Ju et al., 2024; Chen et al., 2024). However, challenges remain in the realm of low-latency streaming zero-shot TTS, where short text chunks are streamed into the model and short audio chunks are streamed out in real-time. Such models are ideal for integration with upstream tasks that emit texts in small chunks such as large language models Achiam et al. (2023); Team et al. (2023) or streaming translation models (Barrault et al., 2023). Addressing these challenges could transform live and interactive communication, paving the way for applications such as low-latency speech-to-speech translation, accent conversion, and responsive voice assistants.

While existing models show promising performance in offline inference, they are not suitable or do not support streaming. When it comes to on-device streaming, autoregressive modeling approaches (Dang et al., 2024; Borsos et al., 2023; Peng et al., 2024) offer an advantage due to the capability of streaming the outputs frame-by-frame. The use of stream-unwary models on streaming inputs involves breaking down the text into short text chunks and condition each generation on previously generated speech, e.g., via prompting. Even when these models are adapted to synthesize from a non-stop text stream, several challenges arise in a low-latency scenario: (1) the fixed text condition during inference complicates seamless updates with arriving text chunks, for example, the generation for a text chunk cannot leverage newly arriving context for lookahead; (2) the speech output must catch up with the leading edge of the text stream, requiring the length of generated speech to adapt to the arrival time of text chunks; and (3) the model must process short text chunks while ensuring smooth transitions between their corresponding generated speech segments. In addition to these technical requirements, the speed of inference remains a challenge for inference on the device, since the transformer decoder has to generate a fairly large number of tokens for a single second of audio.

In this paper, we propose L3Speech with additional capabilities to overcome aforementioned challenges. First, we adopt Mamba, a recently developed and highly capable recurrent architecture for sequence modeling, and are the first to demonstrate its competitiveness against transformer-based

Table 1: Comparisons that highlight the capabilities of our proposed models. Stream-unwary models face numerous challenges when adapting to chunk-level streaming scenarios.

| Capability | Non-Streaming Models | Our Proposed Models |
|---|---|---|
| Non-stop speech streaming | NO support for text streaming in. Texts need to be fixed during generation. Long texts must be segmented into sentences. | allow for a sliding window over long text sequences, retaining only the relevant text in the context for each decoding step. |
| Text-audio stream synchronization | NO support for duration control. Speech may become out-of-sync with the text stream | allow for generating speech that adjusts to keep pace with the arriving text stream. |
| Seamless transitions between short speech chunks | NO support for conditioning the current generation on previous outputs, causing non-smooth transitions and style inconsistencies. Even when prompting with previous outputs, a ramp-up time remains essential. Moreover, these models usually only support chunks as long as a full sentence. | allow for smooth transitions between chunks and maintaining consistency in styles to past chunks without ramp-up time. Our model consistently emits speech frame in near-constant time, independent of the incoming text. It also supports text chunk lengths as short as a single word. |

counterparts at large scale. Mamba maintains an internal state and only takes $O(1)$ complexity to perform a decoding step, thus reducing the inference time compared to transformer-based decoder. We also reduce the memory length for the reference enrollment speech and transcript by compressing them using a transformer-based speech encoder and a byte pair encoding (BPE) tokenizer, respectively. Second, we propose a cross-attention computation method using rotary positional embeddings, enabling a sliding-window approach on the text. This allows the text condition to be updated at any decoding step and facilitates the generation of content beyond the maximum length for which the model was initially trained. Third, we include semantic tokens together with acoustic tokens in the decoding step outputs and propose inference-time semantic guidance to mitigate the misalignment between text and speech. These improvement enables our models to function reliably with low latency in streaming scenarios, particularly when the upstream task outputs long text in short chunks. Table 1 highlights the new capabilities and compares ours with non-streaming models. We conduct experiments to demonstrate that our model perform competitively with state-of-the-art non-streaming models in terms of content accuracy, speaker similarity, and general audio quality. Experimental results on the LibriLight and LibriTTS dataset demonstrate that our models achieve superior speaker similarity and overall audio quality while providing flexibility to balance latency and content accuracy in streaming scenarios. Audio samples are available in supplemental materials.

## 2 RELATED WORKS

Recently, progress in audio and speech generation has focused primarily on the utilization of language models (Borsos et al., 2023; Copet et al., 2024; Wang et al., 2023a; Chen et al., 2024; Casanova et al., 2024) and diffusion models (Tan et al., 2024; Shen et al., 2023; Ju et al., 2024; Le et al., 2024; Bai et al., 2023), with the debate remaining unsettled. Diffusion models demonstrate their potential by directly generate continuous features without relying on an audio codec, offering high content accuracy and inference speed thanks to the non-autoregressive backbone. On the other hand, language models excel in output streaming (Dang et al., 2024), with recent studies (Chen et al., 2024) claiming to achieve human parity on the LibriTTS and VCTK test sets. Both approaches can generate high-quality outputs in non-streaming mode, where the transcript and enrollment speech are available before the generation process starts. Recent works also explore replacing transformer-based decoders with recurrent architectures (Lemerle et al., 2024; Halloran et al., 2024), showing comparable performance at smaller scales.

Most research on streamable TTS emphasizes the adoption of fully autoregressive architectures (Dang et al., 2024; Łajszczak et al., 2024), often overlooking the latency caused by sentence forma-

tion. When it comes to chunk-level streamable TTS systems, Dekel et al. (2024) train a streaming TTS model by distilling from a non-streaming TTS with limited access to future context; however, the architecture does not have a strong zero-shot capability (in fact, it is only demonstrated for a single speaker), and the distillation process only supports one setting for the chunk length and chunk lookahead. Transducers, an architecture known for its streaming advantages in speech recognition, have recently been adapted to the zero-shot TTS task (Kim et al., 2023; Du et al., 2024a), showcasing potential for streaming capabilities. However, these works have not been fully adapted to a true streaming scenario, as they rely on non-autoregressive transformers to generate fine audio tokens and lack a duration control mechanism. Additionally, it remains unconfirmed whether the architecture can scale to sigificantly larger datasets (e.g., from LibriTTS's 600 hours to LibriLight's 60k hours)

Our work demonstrates streaming capabilities similar to those of recent efforts on full-duplex models (Ma et al., 2024; Défossez et al., 2024; Wang et al., 2024), where speech language models can listen and speak simultaneously; however, while those typically focus on improving interruptibility for conversational models, our goal is to synthesize an existing incoming text stream with minimal latency.

## 3 BACKGROUND

### 3.1 AUDIO COMPRESSION WITH RESIDUAL VECTOR QUANTIZATION (RVQ)

An audio tokenizer is crucial when using a language model decoder to generate audio. Usually, the audio tokenizer is an audio codec (Zeghidour et al., 2021; Défossez et al., 2022; Kumar et al., 2024; Jiang et al., 2023; Du et al., 2024b; Siuzdak, 2023) with an encoder, a quantizer, and a decoder. The encoder transforms the audio signal into a latent representation of $T$ time steps $\boldsymbol{z}_1, \boldsymbol{z}_2, ..., \boldsymbol{z}_T$, which is recursively quantized by a sequence of quantizers to produce $Q$ codes $\boldsymbol{c}_i = [c_i^{(1)}, c_i^{(2)}, ..., c_i^{(Q)}]$ for each frame feature $\boldsymbol{z}_i$. Audio tokens can be generated in the same way as language tokens; however, the amount of tokens poses a challenge of high inference time when being predicted sequentially (Borsos et al., 2023). MusicGen (Copet et al., 2024) reduces the number of decoding steps by shifting the codes to predict $Q$ codes in a single step, each of which comes from one in consecutive frames. LiveSpeech (Dang et al., 2024) also applies the shifting techniques; however, $Q$ codes are divided into groups that are modeled independently in parallel. Stack-And-Delay (Le Lan et al., 2024) also processes shifted codes in parallel to find a balance between performance and inference speed.

### 3.2 LINEAR-TIME SEQUENCE MODELING WITH MAMBA

Based on Structured State Space Sequence (S4) models (Gu et al., 2021). In general, it involves a continuous system that maps a sequence $x(t)$ to $y(t)$ through a latent state $h(t)$, defined by four parameters $\boldsymbol{\Delta}, \boldsymbol{A}, \boldsymbol{B}, \boldsymbol{C}$, fomulated as $h'(t) = \boldsymbol{A}h(t) + \boldsymbol{B}x(t), y(t) = \boldsymbol{C}h(t)$. After discretizing with zero-order hold: $\overline{A} = \exp(\Delta \boldsymbol{A}), \overline{B} = (\Delta \boldsymbol{A})^{-1} (\exp(\Delta \boldsymbol{A} - I)) \cdot \Delta \boldsymbol{B}$, the computation becomes $\boldsymbol{h}_t = \overline{\boldsymbol{A}}\boldsymbol{h}_{t-1} + \overline{\boldsymbol{B}}\boldsymbol{x}_t, \boldsymbol{y}_t = \boldsymbol{C}\boldsymbol{h}_t$, which provides a linear recurrence computation for autoregressive inference. The model can also be computed via global convolution for efficient parallelizable training: $y = x * (C\boldsymbol{B}, C\boldsymbol{A}\boldsymbol{B}, \ldots, C\boldsymbol{A}^k\boldsymbol{B}, \ldots)$

Mamba overcomes the linear time-invariance constraint of S4 models, while still maintaining computation efficacy. In particular, the parameters $\boldsymbol{\Delta}, B, C$ are functions of the input, and an efficient hardware-aware implementation is used to replace the global convolution computation.

We adopt Mamba (Gu & Dao, 2023) as the language modeling component in our model to replace transformers in previous work (Wang et al., 2023a). Transformers require attention computation over the past context without any compression, which is computationally inefficient on long sequences, Mamba, on the other hand, summarizes the context into a fixed size state vector via the selection mechanism. We believe that state-space models are the more efficient choice for language modeling of audio tokens since audio tokens are usually long, redundant, and biased towards recency. Rather than storing the entire past generation, a compressed state could provide enough information to ensure smooth frame transition and semantic coherence.

# 4  L3SPEECH

In this section, we present L3Speech, our zero-shot TTS model with the streaming capability. The model processes a continuous text stream and outputs codec codes on a frame-by-frame basis. The transcript is delivered in short text chunks, taking into account the timing of arrival. Following the overall architecture of LiveSpeech Dang et al. (2024), our model contains three main components: a speech encoder that encodes enrollment speech, a text tokenizer and embedder that embed text chunks, and an autoregressive decoder.

The speech encoder is a transformer-based encoder that converts enrollment speech of arbitrary length into a $N_s$ fixed-length sequence of embeddings by appending $N_s$ zero vectors to the inputs and use the outputs of these positions as the encoder outputs. ~~The embeddings can remain unchanged or be updated any time during streaming~~. The primary objective of this encoder is to extract a significantly compressed representation of the entire speech, thereby accelerating the decoding time. The text tokenizer extracts token indices and the text embedder outputs a sequence of token embeddings. We employ the byte pair encoding (BPE) tokenizer from Whisper (Radford et al., 2023) for its extensive coverage and compatibility with upstream Whisper model outputs. We call these tokens word tokens, although some of them do not represent complete words. An end-of-stream token (EOS) is used to signal the end of generation. For the decoder, we employ Mamba (Gu & Dao, 2023) as an alternative to transformers typically used in related works. In addition to offering competitive performance with a linear-time decoding approach compared to transformers, we posit that speech generation necessitates access not to all tokens in history, but only to a continuously updated state. The decoder integrates information from speech and text embeddings through cross-attention.

To facilitate streaming, we maintain in memory only the current and its neighboring text chunks, updating them continuously as decoding progresses or new chunks arrive. However, the model is trained using fixed transcript with a maximum length, resulting in a disparity between training and inference time. We address the challenges as follows. Section 4.1 details our approach to enable dynamic text by assigning positional indices aligned with speech to each word token, and employing rotary positional embeddings when computing cross-attention between speech and text. Section 4.2 introduces a method to prevent misalignment by leveraging monotonic semantic guidance from the transcript.

## 4.1  TEXT-SPEECH CROSS-ATTENTION WITH ROTARY POSITIONAL EMBEDDING

Let $S_1, S_2, S_3 \ldots$ be chunks from a text stream, where each chunk $S_i$ is a sequence of text tokens $w_i^1, \ldots, w_i^{|S_i|}$. Assume that the number of tokens in $S_i$ is bounded by $l_{\min} \leq |S_i| \leq l_{\max}$. We introduce an additional input $t_i$, which denotes the time between the arrival of chunk $i-1$ and $i$. To facilitate streaming, the speech for the $i$-th chunk has a duration of approximately $t_i$ in the number of frames. This assumption ensures that a new chunk is not delayed by the completion status of the previous chunk. For example, starting from the time 0, if the first chunk arrives at the time $t_1$, but takes $t_1' > t_1$ to complete, the playback of the second chunk will be delayed by the audio-text duration gap $t_1' - t_1$. Let $\tau_i = \sum_{i' \leq i} t_{i'}$, which is the time step at which the chunk $S_i$ starts.

During inference, we aim to add context when new chunks arrive and remove context when it is no longer necessary for generation. However, during training, the context for a single sample is typically fixed for all decoding steps. We adopt a straightforward approach by granting full access to context during training but retaining only certain relevant chunks for each decoding step during inference. We introduce two inference-time hyper-parameters in our system: the maximum number of past chunks included in the cross attention memory, denoted as $n_p$, and the maximum number of future chunks included in the cross attention memory, denoted as $n_f$. In particular, the decoder can attend to $n_p + n_f + 1$ chunks, $(S_{i-n_p}, t_{i-n_p}), \ldots, (S_i, t_i), \ldots, (S_{i+n_f}, t_{i+n_f})$, to generate speech for $S_i$ in $t_i$ steps. When $n_f = 0$, the system starts generating immediately after a text chunk arrives. When $n_f > 0$, the system delays generation until chunk $S_{i+n_f}$ arrives.

**Positional indices based on arrial time**  For each word token embedding, we assign a position index to it: word tokens $w_i^1, w_i^2, \ldots, w_i^{|S_i|}$ from chunk $S_i$ are assigned with position indices $\tau_i, \tau_i + 1, \ldots \tau_i + |S_i| - 1$. Figure 2 illustrates this assignment.

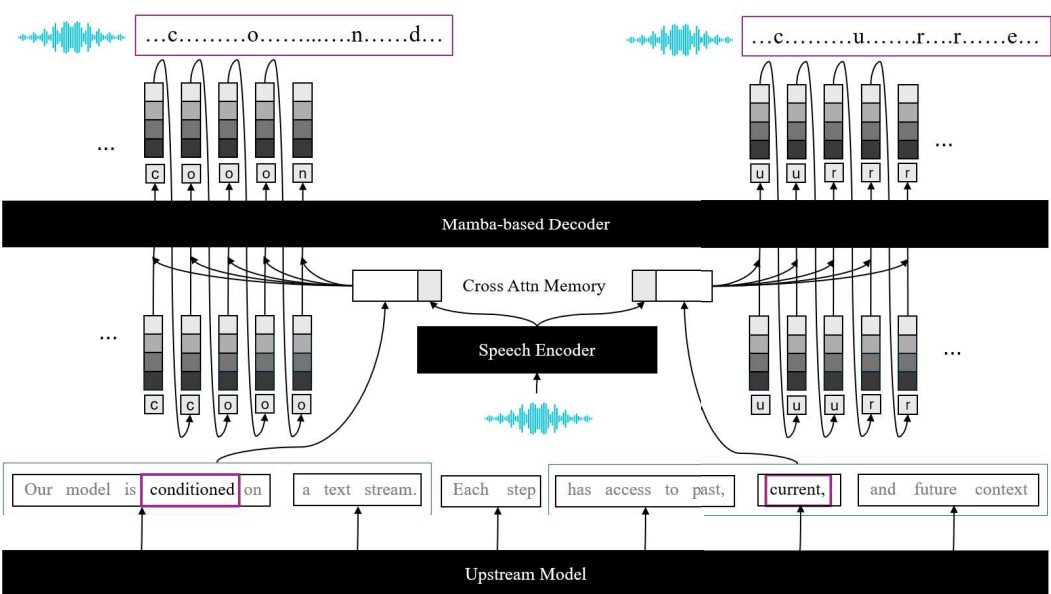

Figure 1: L3Speech general architecture. An upstream model generates text continuously in small chunks, while our model synthesizes speech, aiming to keep pace with the most recent chunk. Besides enrollment speech embeddings, each decoding step has access to a section of the text stream, including some past and future chunks.

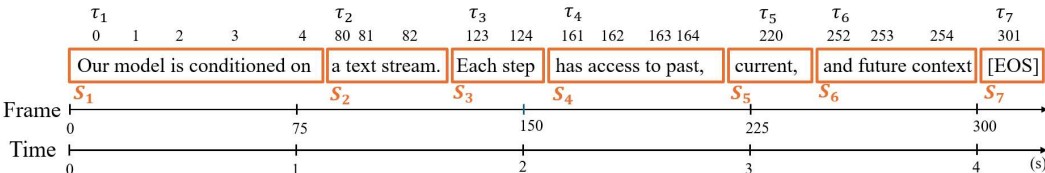

Figure 2: For each word token in a chunk $i$, we assign a position index such that the first word has the index of the frame when the chunk arrives $\tau_i$, and subsequent words have incremental indices $\tau_i + 1, \ldots$

**Cross-attention computation** Cross-attention scores are computed with the enrollment speech features and the word token embeddings. The enrollment speech features are position-agnostic, while the word token embeddings are coupled with positional indices. Let $c(t)$ be the chunk index at the time step $t$, $p(t) = \max\{0, c(t) - n_p\}$ and $f(t) = \min\{|S|, c(t) + n_f\}$ be the first and the last chunks in the memory for the time step $t$. The attention keys for text at the time step $t$ are expressed as: $\boldsymbol{K}_t^{(\text{txt})} = \left[ \boldsymbol{K}_{p(t)}; \ldots; \boldsymbol{K}_{c(t)}; \ldots; \boldsymbol{K}_{f(t)} \right] \in \mathbb{R}^{\left( \sum_{p(t) \leq i \leq f(t)} |S_i| \right) \times d_k}$. The attention keys for enrollment features are denoted by $\boldsymbol{K}_t^{(\text{enr})}$. Let $\mathcal{T}_t$ be the position indices assigned for each key in $\boldsymbol{K}_t^{(\text{txt})}$. With our positional index assignment, $\mathcal{T}_t = \left[ \tau_{p(t)}, \ldots, \tau_{f(t)} + |S_{f(t)}| - 1 \right]$. For each Mamba layer, let $q_t$ be the layer input at the time step $t$. The cross attention is computed as follows:

$$\boldsymbol{a}_t = \text{Softmax}\left( \left[ \frac{\boldsymbol{q}_t \boldsymbol{K}^{(\text{enr})T}}{\sqrt{d_k}}; \frac{\text{RoPE}(\boldsymbol{q}_t, t)\text{RoPE}(\boldsymbol{K}_t^{(\text{txt})T}, \mathcal{T}_t)}{\sqrt{d_k}} \right] \right), \tag{1}$$

where $\text{RoPE}(\boldsymbol{K}, \mathcal{T})$ rotates the key matrix $\boldsymbol{K}$ given indices $\mathcal{T}$. The attention-weighted sum of cross-attention values is integrated with the output of the Mamba layer and given to the subsequent layer.

## 4.2 INFERENCE-TIME SEMANTIC GUIDANCE

Autoregressive TTS decoding suffers from the problem of misalignment, resulting in missing, transpositioning, or repeating content (Wang et al., 2023a). To address this issue, we propose providing guidance during inference based on the conditioned transcript.

**Training** During training, we use time-aligned graphemes as an additional codebook, placed before the first acoustic codebook. Time-aligned graphemes can be obtained from the output of a CTC-based ASR model. Since this output includes plenty of blank tokens, making it sparse in terms of non-blank tokens, we replace each blank token with the first non-blank tokens to the right of the sequence. As an example, grapheme sequence "abc" can have a time aligned grapheme sequence of "aa␣␣bbbb␣␣␣␣␣␣cc␣␣", which will be processed to become "aaaaabbbbbbbbbbcc␣␣". Given that there are 75 acoustic tokens per second, while most CTC models generate only 50 tokens per second, we interpolate this sequence via its one-hot vectors to align with the number of decoding steps.

**Inference** During inference, we use the previously generated graphemes $G_{t-1} = [g_1, \ldots, g_{t-1}]$ and the transcript to guide the decoding of the next grapheme $g_t$. Let $\boldsymbol{p}_t^{(g)} = \left[p_{t,1}^{(g)}, \ldots, p_{t,N_g}^{(g)}\right] \in \mathbb{R}^{N_g}$ be the probability distribution predicted for the next grapheme, where $N_g$ is the number of graphemes. We infer a set of guiding tokens $T_{\text{guiding}}$ from the current grapheme sequence and the transcript by determining the prefix of the transcript that matches most closely with $C_{t-1}^{(g)}$. Guiding tokens are either the last token in the prefix (staying) or the next token following the prefix (moving forward). We also infer a set of top-k tokens $T_{\text{top-k}}$ by taking graphemes with highest probability. The next grapheme is sampled from $T_{\text{guiding}} \cup T_{\text{top-k}}$ with a reweighted probability determined by upscaling the probability of guiding graphemes by $(1 + \lambda)$ and renormalizing. When $\lambda = 0$, *no guidance* is provided. When $\lambda \to \infty$, we call it *hard guidance* when the next grapheme is only chosen from guiding graphemes. When $0 < \lambda \ll \infty$, we call it *soft guidance* where the guiding graphemes are factored in the choice of the next grapheme. In short, we identify a set of graphemes such as if we append one of those to the generated time-align grapheme sequence, this new grapheme sequence has the least CER score to a prefix sequence of the transcript. Hard guidance expects the grapheme sequence to exactly follow the transcript, while soft guidance allows mistakes in the process. Algorithm 1 illustrates the sampling process with semantic guidance.

---

**Algorithm 1:** Autoregressive decoding with semantic guidance

**Data:** Target transcript $\bar{G}_t = [\bar{g}_1, \bar{g}_2, \ldots \bar{g}_{|\bar{G}_t|}]$. Previous decoded graphemes $G_{t-1} = [g_1, g_2, \ldots g_{t-1}]$. Softmax probability of the next grapheme $\boldsymbol{p}_t^{(g)} = \left[p_{t,1}^{(g)}, \ldots, p_{t,N_g}^{(g)}\right]$. Guiding coefficient $\lambda$. Number of graphemes $N_g$

**Result:** Next grapheme $g_t \in [1, ..., N_g]$

$\tilde{G}_{t-1} := \text{CTCDecode}(G_{t-1})$ ;          // remove repetitive/non-char tokens

$s_{\text{CER}} := \min_i \left\{ \text{CER}(\tilde{G}_{t-1}, \bar{G}_t[: i]) \right\}$ ;    // the best Character Error Rate

$T_{\text{guiding}} := \{\}$

**for** $i \in [1, \ldots, |\bar{G}_t|]$ **do**
    **if** *CER*$(\tilde{G}_{t-1}, \bar{G}_t[: i]) = s_{CER}$ **then**
        $T_{\text{guiding}} := T_{\text{guiding}} \cup \{\bar{g}_i, \bar{g}_{i+1}\}$
    **end**
**end**

$T_{\text{top-k}} := \left\{ k \mid p_{t,k}^{(g)} \in \text{TopK}\left(\boldsymbol{p}_t^{(g)}\right) \text{ and } k \notin T_{\text{guiding}} \right\}$

$\boldsymbol{p}_{t,k}^{(g)} := \begin{cases} \boldsymbol{p}_{t,k}^{(g)} & \text{if } k \in T_{\text{top-k}} \\ \boldsymbol{p}_{t,k}^{(g)} \times (1 + \lambda) & \text{if } k \in T_{\text{guiding}} \\ 0 & \text{otherwise} \end{cases}$ ;        // reweighing probabilities

$\boldsymbol{p}_t^{(g)} := \frac{\boldsymbol{p}_t^{(g)}}{\sum \boldsymbol{p}_t^{(g)}}$ ;                              // normalize probabilities

$g_t \sim \text{TopKSampling}(\boldsymbol{p}_t^{(g)}, k)$

---

While explicitly generating semantic tokens as a transitional "language" between the transcript and acoustic tokens has been proposed (Borsos et al., 2023; Kharitonov et al., 2023), semantic tokens only serve as the condition to generate acoustic tokens. We take a further step to use the transcript to guide the decoding process in inference time with flexibility.

In this paper, we focus on English as the target language, selecting graphemes as semantic tokens. However, alternative units could be utilized to accommodate a wider spectrum of languages and applications. It is important to choose a unit that allows the transcript to be used to refine candidates in the decoded sequence. Therefore, both graphemes and phonemes are feasible, whereas self-supervised semantic tokens may present certain challenges.

## 5    EXPERIMENTS

### 5.1    DATASETS

For training, we use LibriLight (Kahn et al., 2020), a 60k hour corpus of unlabelled speech for training. We use Whisper v3 (large) (Radford et al., 2023) and wav2vec 2.0 (base) (Baevski et al., 2020) to extract the transcript and its word alignment to speech from each training sample. We observe that while Whisper v3 generally produces transcripts with lower error rates and support for punctuation and abbreviations, it occasionally fails catastrophically. Therefore, we use wav2vec 2.0 transcripts and alignments to filter out poor-quality training samples. Specifically, a sample is discarded if the character error rate (CER) between the transcripts from the two models exceeds 0.1 or if the alignments do not match. Each sample is less than 10 seconds in duration, and an enrollment speech of less than 5 seconds from the same speaker is also extracted.

For evaluation, we utilize samples from the test-clean set of LibriTTS. The test set is filtered and divided into two subsets: (1) target speech samples of 3-10 seconds in duration (totaling 2,288 samples, with an average duration of 5.8 seconds), and (2) target speech samples longer than 10 seconds (totaling 1,002 samples, with an average duration of 14.7 seconds).

For both training and evaluation, we simulate a text stream by randomly dividing the transcript into chunks of 2 to 4 word tokens. The alignments from Whisper v3 are used to infer the arrival time of these chunks. The final chunk contains an end-of-stream (EOS) token, with its time set to the duration of the corresponding speech.

### 5.2    MODEL

We report results using popular baseline models such as: YourTTS (Casanova et al., 2022), XTTS v2 (Casanova et al., 2024), MetaVoice (MetaVoice Team, 2024), SpeechX (Wang et al., 2023b), and LiveSpeech (Dang et al., 2024). All model code and checkpoints are either public (YourTTS, XTTS v2, MetaVoice) or provided by the authors (Speech X, LiveSpeech).

In our model, the speech encoder is a 6-layer 8-head transformer encoder with a hidden dimension of 1024. We prepend 64 empty features to the enrollment speech features to extract a vector sequence of length 64 representing the speech. The Mamba-based decoder consists of 12 layers with a hidden dimension of 1536. The transcript is tokenized with a vocabulary of 51,866 word tokens the same as Whisper (Radford et al., 2023). Following LiveSpeech (Dang et al., 2024), the first 6 layers are shared to model all codebooks, and the last 6 layers divide codebooks into 4 groups of 4, 4, 4, 5 codebooks, respectively, which are modeled separately. We also apply a weight based on the codebook prediction performance with $\lambda_{cb} = 0.1$ (Dang et al., 2024). The cross attention has 16 heads with a hidden dimension of 1536. The maximum length for the cross attention memory is 64 + 75, where 64 features belong to enrollment speech and 75 features belong to maximum 75 word tokens in the transcript. Our audio codec, speech encoder, and decoder have 110M, 77M, and 671M parameters, respectively.

### 5.3    TRAINING & INFERENCE

**Training** We use Encodec to extract acoustic codes at the bit rate of 12kbps or 16 codes/frame and 75 frames/second. Since Encodec is also trained on general audio and music, we train a new decoder specialized in speech on the LibriLight dataset. The model is trained for 2M steps with batch size

32 on 4 A100 GPUs. We employ a learning rate of $5 \times 10^{-4}$ with 200k warm up steps (Smith & Topin, 2019).

**Inference** We perform two modes of inference: offline inference for 3-10s speech and online inference for speech longer than 10s. For offline inference, all past text chunks ($n_p = \infty$), the current, and $n_f = 2$ future text chunks are accessible at each decoding step. For online inference, we slide a window over seven chunks, including $n_p = 4$ before and $n_f = 2$ after the current chunk being generated. Since each chunk has 2-4 words, our system delays 4-8 words after a chunk arrives until its speech can be streamed. If not specified otherwise, we use semantic guidance with $\lambda = 1$.

### 5.4 EVALUATION METRICS

We evaluate our models in terms of objective and subjective metrics.

**Objective Metrics** In terms of content accuracy, we report the Character Error Rate (CER) score with the transcript obtained through the wav2vec2 base model (Baevski et al., 2020) and Word Error Rate (WER) score with the transcript obtained via the Whisper v3 model (Radford et al., 2023). While Whisper v3 is a stronger model that may give us scores closer to human transcripts, wav2vec2 is expected to give more penalty to pronunciation mistakes. In terms of speaker similarity, we report the cosine similarity scores between the generated and the enrollment speaker embeddings using the ECAPA-TDNN model trained on Vox-Celeb (Desplanques et al., 2020). In terms of general speech quality, we report DNSMOS scores (Reddy et al., 2022).

**Subjective Metrics** We measure Mean Opinion Score in terms of speaker similarity (SMOS) and naturalness (NMOS). For SMOS, we ask each subject to rate the speaker similarity of the enrollment speech and the speech to be evaluated in a scale of 5. For NMOS, we ask each subject to rate the naturalness of the speech in a scale of 5. For each sample, we allow subjects to adjust scores after listening to all audio clips, facilitating relative comparisons between different models. There are 30 short and 30 long samples, each of which is rated by an average of approximately 5 and 3 subjects, respectively.

### 5.5 RESULTS

The results are reported in Table 2. In terms of CER / WER scores, we are only behind the XTTS v2 baseline, which is trained on a massive amount of internal and public data. Some models have been found to achieve CER and WER scores that surpass even ground-truth samples, indicating that achieving these scores might involve trading off real speech characteristics for improved CER and WER metrics (Peng et al., 2024) (e.g., emphasizing clean audio over audio that resembles enrollment speech). Our model achieves the highest SS score. In terms of subjective metrics, our model outperforms all baselines in both SMOS and NMOS scores, where more significant improvements are also observed for long inputs in the streaming mode.

Additionally, we observe a larger gain compared to baselines in speaker similarity when using long utterances, both in objective and subjective metrics. In particular, for the short duration test set, we gain +1.4/+0.3 in objective/subjective scores compared to +2.8/+0.5 on the long duration test set, compared to XTTS v2. This suggests that our model is not affected by long duration as much as other baselines that have to split the text to fit in the maximum context length.

**Real Time Factor (RTF)** Our real-time factor (RTF) is 1.6 without the CUDA-optimized kernel (used to obtain our results) and 0.83 with the CUDA-optimized kernel, excluding the time required for computing enrollment speaker embeddings. Reducing the padded context length to 25 and the number of speaker embedding vectors to 1 further lowers the RTF to 0.77. On the same hardware, our best baseline, XTTS-v2, achieves an RTF of 0.43. It is worth noting that we have not employed inference optimization techniques, and the RTF may have less impact on latency compared to the lookahead in a streaming scenario.

**Latency** In terms of latency, we ignore the time-to-first-frame, assuming that fully autoregressive architectures can start producing speech right after the text becomes available, and consider only latency incurred by the lookahead. In this regard, most baseline models incur a latency of one sentence length. Our model can achieve latency as low as 6 words, comprising one current chunk

Table 2: Comparison of our model to the baselines. Each metric is reported with 3-10s / longer than 10s for the target speech. We do not report results of samples longer than 10s for SpeechX, MetaVoice, and LiveSpeech since some samples exceed their maximum context length. For YourTTS and XTTS v2, long transcript is split by Coqui-TTS (Eren & The Coqui TTS Team, 2021) into smaller ones, which are synthesized separately. Only our model generates speech for all samples in one shot.

| Model | CER | WER | SS | O-MOS | SMOS | NMOS |
|---|---|---|---|---|---|---|
| Ground-truth | 1.6 / 1.4 | 0.6 / 0.7 | 75.3 / 83.9 | 3.9 / 4.0 | 3.8 / 4.0 | 3.7 / 4.0 |
| Ground-truth (compressed) | 1.7 / 1.5 | 1.0 / 1.2 | 71.1 / 79.3 | 3.9 / 4.0 | 3.5 / 3.9 | 3.5 / 3.9 |
| YourTTS (Casanova et al., 2022) | 3.8 / 3.3 | 4.3 / 3.6 | 48.6 / 55.2 | 3.8 / 3.9 | 2.6 / 2.3 | 2.5 / 1.9 |
| XTTS v2 (Casanova et al., 2024) | **1.9 / 2.2** | **1.3 / 1.9** | 60.3 / 64.8 | **4.0 / 4.0** | 3.1 / 2.9 | 3.0 / 3.1 |
| SpeechX (Wang et al., 2023b) | 3.8 / — | 4.4 / — | 57.6 / — | 3.8 / — | 2.3 / — | 2.1 / — |
| MetaVoice (MetaVoice Team, 2024) | 4.7 / — | 4.1 / — | 56.2 / — | 3.7 / — | 2.2 / — | 2.2 / — |
| LiveSpeech (Dang et al., 2024) | 3.3 / — | 6.0 / — | 59.3 / — | 3.8 / — | 2.8 / — | 2.5 / — |
| L3Speech (ours) | 2.7 / 3.0 | 3.1 / 4.1 | **61.7 / 67.6** | 3.9 / **4.0** | **3.4 / 3.4** | **3.2 / 3.3** |

Table 3: Ablation study on the necessity of semantic tokens and semantic guidance.

| Model | WER | SS |
|---|---|---|
| With sem guidance | **3.1 / 4.1** | **61.7 / 67.6** |
| W/o sem guidance | 6.7 / 5.6 | 61.3 / 67.4 |
| W/o sem tokens | 7.3 / 13.4 | 60.6 / 67.0 |

Table 4: Results when each sample is generated $N$ times and selected based on CER or probability scores.

| Model | WER | SS |
|---|---|---|
| 1-time | 3.1 / 4.1 | 61.7 / 67.6 |
| 2-time (CER based) | 2.3 / 3.6 | 61.8 / 67.6 |
| 5-time (CER based) | **2.0 / 3.1** | 61.8 / 67.6 |
| 5-time (prob based) | 2.1 / 3.5 | **61.9 / 68.6** |

and two lookahead chunks, with each chunk containing as few as two words. Furthermore, our audio-text stream synchronization ensures that playing the audio for a new chunk is not delayed by the completion status of the previous chunk.

## 5.6 Ablation Study & Analysis

**The importance of semantic tokens and semantic guidance** We conduct an ablation study when the model does not generate semantic tokens and when they are generated but semantic guidance is not used. Table 3 shows the results. By including semantic tokens in each step, we are able to obtain significant gains in the WER score, especially for long speech where error propagation is more problematic. Semantic guidance also shows considerable effect on the content accuracy, with 53% improvement in offline scenario and 27% improvement in online scenario.

**N-time sampling** Existing studies (Chen et al., 2024; Shen et al., 2023; Peng et al., 2024) utilize simple heuristics to select the output from multiple generated outputs; these heuristics range from length-based to metric-based criteria. By incorporating grapheme tokens in our model outputs, transcripts and CER scores of generated speeches become available without the need for an ASR system. Table 4 illustrates the improvement gains for N-time sampling and compares them with a probability-based criterion, where outputs are selected based on the cumulative probability of the entire sequence of graphemes. Although the probability-based criterion does not guarantee optimal CER scores, it can select the highest in overall probability among those with the same CER scores, thereby resulting in an improved SS score (+1.0). It is important to note that N-time sampling is applicable only for offline inference.

**Effects of the text chunk length** The chunk lengths depend on the upstream task. When only a small local context is required to infer the text (e.g., transcribing), we expect short chunks and lower latency. When the inference of the text requires more global context (e.g., translating), longer chunks are usually needed for better accuracy. For the same transcript, we investigate how different chunking situations affect the quality of the generation. Table 5 shows results for different ranges

Table 5: Results for different text chunk minimum ($l_{min}$ words) and maximum ($l_{max}$ words) lengths

| $l_{min}$ | $l_{max}$ | WER | SS |
|---|---|---|---|
| 1 | 1 | 40.7 / 73.7 | 58.2 / 65.3 |
| 1 | 3 | 6.8 / 8.1 | 61.6 / 69.5 |
| 2 | 2 | 3.4 / 4.8 | 60.6 / 69.0 |
| 2 | 4 | 4.0 / **3.9** | **62.2 / 70.1** |
| 3 | 7 | **3.6** / 4.5 | 62.1 / 69.8 |

Table 6: Results for different text chunk lookback ($n_p$ chunks) and lookahead ($n_f$ chunks)

| $n_p$ | $n_f$ | WER | SS |
|---|---|---|---|
| 1 | 1 | 23.5 / 15.6 | 61.3 / 68.2 |
| 10 | 1 | 7.5 / 8.5 | 61.5 / 68.9 |
| 2 | 2 | 3.3 / 4.6 | 61.1 / 69.5 |
| 10 | 2 | 3.8 / 3.7 | **62.3** / 69.8 |
| 10 | 4 | **3.0 / 3.3** | 61.3 / **69.9** |

Table 7: Robustness of our model on perturbed chunk boundaries. Results are shown for the online scenario.

| $\Delta\tau$ | WER | SS | O-MOS |
|---|---|---|---|
| 0.0 | 2.7 | 70.5 | 3.99 |
| 0.2 | 3.6 | 69.5 | 3.95 |
| 0.4 | 5.3 | 68.5 | 4.00 |

$[l_{min}, l_{max}]$. Our model perform poorly in WER score when each chunk has only one word token, hinting that further fine-tuning is required for this extreme scenario. We provide results on streaming aware training in the Appendix A.5, where WER scores are significantly improved even when each chunk has only one word. WER score significantly improves when we increase the range to $[1, 3]$ or $[2, 2]$, and continues to improve as the chunk length increases.

**Effects of the number of text chunks** We investigate the impact of modifying the extent of access to preceding ($n_p$) and succeeding ($n_f$) text chunks on the content fidelity and the audio quality of synthesized speeches. The model exhibits suboptimal performance when constrained to only a single chunk from both preceding and succeeding contexts; however, its efficacy improves with the expansion of access to prior chunks. When the model is allowed to see more of future chunks ($n_f > 1$), its performance significantly improves. We also observe an improvement in SS scores when extending the number of past chunks from 2 to 10, suggesting that access to a longer text history enhances certain aspects of voice style.

**Effects of perturbation on chunk boundaries** In some applications, the text chunk duration inferred by the text chunk arrival time does not exactly match the audio chunk duration. For example, in the speech translation task, the arrival time indicates the duration in the source language, whereas the ideal duration should correspond to the target language; however, they should be closely aligned. We present experiments in scenarios with perturbed chunk boundaries. In particular, for each 2-4 word chunks as in the main experiment, we randomly uniform noise in $[-\Delta\tau; \Delta\tau]$ to simulate this scenario. Table 7 presents our results, showing some degradation when the chunk boundaries are noisy. Although the degradation is slight, given the maximum random time of 0.4 seconds added to chunk boundaries that are approximately 1 second in duration, this also suggests that additional fine-tuning may be necessary to mitigate the impact, depending on the type of the upstream task.

# 6 CONCLUSION & SOCIETAL IMPACT

We introduced L3Speech, a zero-shot text-to-speech (TTS) model capable of real-time audio synthesis from continuous textual input. Our model supports real-time applications by continuously streaming short text chunks into the model while producing audio chunks at a constant pace. Given its ability to synthesize speech for any voice, there are concerns regarding possible misuse.

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
