# OpenReview forum: "Zero-Shot Text-to-Speech from Continuous Text Streams"
_ICLR.cc/2025/Conference — Submitted to ICLR 2025_

### Official Review · Reviewer_aFLh · 2024-10-26

**Soundness:** 1
**Presentation:** 3
**Contribution:** 3
**Rating:** 3
**Confidence:** 5

**Summary:**

This study introduces a low-latency streaming zero-shot TTS system with several key advancements: 1) the adoption of Mamba architecture for linear-time decoding, 2) the incorporation of rotary positional embeddings within the cross-attention module to manage infinite text streams, and 3) the implementation of semantic guidance to enhance system robustness. The proposed system achieves comparable performance in both offline and online inference settings.

**Strengths:**

- This work presents a novel scenario for zero-shot TTS systems and highlights its non-trivial challenges.

- This work validates the feasibility of Mamba, a linear-time decoding sequence modeling architecture, for zero-shot TTS tasks, providing insights that may inspire future research.

- The proposed streaming zero-shot TTS model achieves comparable performance in a simluated online inference setting.

**Weaknesses:**

- Some key motivations in this paper, such as those mentioned in Table 1, haven't been thoroughly validated. Specifically:

  - The authors should consider adding experiments that test the model's robustness with sufficiently long text streams to support its claim of "infinitely long speech streaming" capabilities. Additionally, it would be beneficial for the authors to design experiments that demonstrate the advantages of modeling entire text streams compared to synthesizing them sentence by sentence.

  - The authors should compare the performance gap between complete texts and incomplete streaming texts to verify the model's ability to make "seamless transitions between short speech chunks".

  - Furthermore, regarding the "text-audio stream synchronization" capability, there seems to be a lack of clear explanation and corresponding experiments.

  - In addition, the paper positions itself as a low-latency streaming zero-shot TTS system, yet it falls short of providing a latency comparison.

- Some experimental details are missing, which could affect the reproducibility of the proposed system. Please refer to the Questions.

**Questions:**

- The paper states, "The embeddings can remain unchanged or be updated any time during streaming." on L191. Could you elaborate on the mechanisms for updating the enrollment speech embeddings? Furthermore, were these embeddings actually updated in the experiments conducted?

- Could you provide statistical information for the reference enrollment speech? From experience, the duration of the reference enrollment speech greatly influences the speaker similarity.

---

> ### Author Response · Authors · 2024-11-16
> **Thank you for your valuable feedback. We believe important details may have been overlooked.**
>
> Thank you very much for your valuable feedback. While we greatly appreciate the constructive feedback and agree that the paper could benefit from a clearer explanation, we believe important details have been overlooked.
>
> ## Weaknesses
>
> > *"adding experiments that test the model's robustness with sufficiently long text streams"*
>
> In our experiments, we demonstrated that our model can operate *beyond the maximum context length that it was trained on*. Our experiments use a filtered set of 10-20s samples (avg: 14.7s), while our model is trained on maximum 10s. We believe that effective shows that our model can handle infinitely long text stream by sliding a window, since our model *does not have access to the absolute position of text tokens* (in the same way as RoPE or a CNN/RNN network). We will, however, include a few longer samples in the attachment for qualitative listening.
>
> > *"design experiments that demonstrate the advantages of modeling entire text streams compared to synthesizing them sentence by sentence."*
>
> We believe that this is our main result. We provided a comparison between our chunk-based inference with limited lookahead (online setting) vs baselines with full sentence available. It is worth noting that our approach has an advantage of a few word latency while sentence-by-sentence inference has one sentence latency.
> > *"compare the performance gap between complete texts and incomplete streaming texts to verify the model's ability to make "seamless transitions between short speech chunks"."*
>
> We believe that this is a part of our main results. In the streaming setting (10-20s), our model performs inference with a limited lookahead, continuously updated with new chunks from the text stream as generation progresses, where in extreme cases each chunk has only one or two words (Table 5, 6, 11-14). Although we do not provide metrics to explicitly measure seamless transitions, the fact that our model performs well in this setting suggests it handles short arriving chunks effectively. You can refer to our included samples for a qualitative listening. Since existing models do not allow updating the text during generation, independently generating these chunks would certainly result in non smooth transitions.
> > *regarding the "text-audio stream synchronization" capability, there seems to be a lack of clear explanation and corresponding experiments.*
>
> - In Section 4.1, we describe our output requirement: "...the speech for the $i$-th chunk has a duration of $t_i$...". We agree that we should explain more how it enables text-audio stream synchronization. For example, starting from the time $0$, if the first chunk arrives at the time $t_1$, but needs $t'_1 > t_1$ to play, the playback of the second chunk will be delayed by the audio-text duration gap $t'_1-t_1$. This can accumulate progressively throughout the generation process.
> - We visualize the attention map in Figure 3 and Figure 4 in the Appendix. In the first layer, we see high attention scores on chunk boundaries forming multiple short segments. In later layers, we see attention scores form a continuous line, suggesting that the audio chunk "stretches out" to keep pace with the arrival of the next text chunk.
>
> > *...yet it falls short of providing a latency comparison.*
>
> Our latency can be inferred from the paper, but we agree that we did not highlight it enough.
> - We assume models can produce speech immediately after the text become available - not all baselines support this but fully autoregressive architectures like ours and LiveSpeech do.
> - Our latency is measured by the lookahead, not milliseconds - where our model reported in Table 2 supports 6 word latency (2-word chunk + 2 chunk lookahead). Our model reported in Table 5, 6 and 11-14 (Appendix) can achieve latency as low as 2 word with slight degradation (Table 14 - Appendix). For baselines, we may assume that they need a full sentence to work normally. Additionally, our audio-text stream synchronization ensures that there will not be accumulated delay from previous chunks, where existing models do not have a similar mechanism.
>
> We will make these points clearer in the updated manuscript at the end of the rebuttal period.
>
> ## Questions
> > *The embeddings can remain unchanged or be updated any time during streaming*
>
> Replacing the embeddings is out of scope of this paper, but will be helpful for certain applications (e.g., prosody aware TTS). We will remove the sentence to avoid confusion.
> > *statistical information for the reference enrollment speech?*
>
> Our reference enrollment is 3-5s (Section 5.1). All baseline models are given the same pairs of (enrollment speech, reference text). We believe effects on the SS is the same for all evaluated models.
>
> By responding early, we look forward to hearing from you to confirm if these points have been addressed satisfactorily!

---

> ### Comment · Reviewer_aFLh · 2024-11-21
> **Thanks for your response!**
>
> Thanks for your response！
>
> > "adding experiments that test the model's robustness with sufficiently long text streams"
>
> The claim of "infinite long speech streaming" is an overstatement, as testing on samples limited to the 10-20s range does not adequately validate this claim. It is advisable to either amend this statement or present experimental results on significantly longer samples.
>
> > "design experiments that demonstrate the advantages of modeling entire text streams compared to synthesizing them sentence by sentence."
>
> Given the variations in experimental setups, including the use of different models and datasets, the superior performance over the baseline models should not be entirely attributed to the comparison between modeling continuous text streams and the sentence-by-sentence synthesis approach. It is advisable to conduct an ablation experiment to more compellingly substantiate the necessatity of modeling the whole context.
>
> > ...yet it falls short of providing a latency comparison.
>
> Relying solely on lookahead for assessing latency might not yield a fair comparison, especially when the proposed method has around 800M parameters, substantially more than some of the baseline models. To achieve a more fair evaluation, it is advisable to include a RTF metric in the analysis.

---

> ### Author Response · Authors · 2024-11-26
> **Thank you for your response!**
>
> Thank you for your response, we truly appreciate your time! We provide additional experiment results and discussions as below:
>
> > "adding experiments that test the model's robustness with sufficiently long text streams"
>
> We want to focus on the fact that our model handles incoming text chunks without having to reset the context whenever it reaches the maximum context length, rather than claiming the capability to operate on infinite text streams, since most TTS models can be applied sequentially to an arbitrarily long body of text. We however agree that using the word "infinite" can lead to unnecessary confusion. We will change our claim to "non-stopping generation", implying that we do not have to break long inputs down to independent rounds of inference.
>
> To demonstrate this better, we have included long samples in the attachments. These utterances are obtained by repeating several >20s utterances 3 times to create >60s samples. We do not expect most metrics to work reliably in this setting; therefore, we rely on qualitative listening and observe no significant degradation over time. We highlight that our model does not have access to absolute positions of the text chunks; and in contrast to models that rely on prompting which may suffer from degradation over time, we always have enrollment speech accessible via cross attention.
>
> We will add these samples to attached supplementary materials together with the updated manuscript by the deadline Nov 27.
>
> > "Given the variations in experimental setups, including the use of different models and datasets, the superior performance over the baseline models should not be entirely attributed to the comparison between modeling continuous text streams and the sentence-by-sentence synthesis approach. It is advisable to conduct an ablation experiment to more compellingly substantiate the necessatity of modeling the whole context."
>
> We hope you can elaborate on experiments that would be helpful. The main purpose of not generating sentence-by-sentence is latency, since there are upstream tasks that can produce short text chunks (e.g., voice-text-voice conversion, speech translation - such as SeamlessM4T) and existing models are not trained to handle short text chunks. You may refer to Table 5 and 6, which show that it is usually better to have longer lookaheads. We believe that models work better when the full sentence is available; however, our model offers a way to start generating without a full sentence, with reasonable trade-off in performance.
>
> > "To achieve a more fair evaluation, it is advisable to include a RTF metric in the analysis."
>
> Our real-time factor is 1.6 without CUDA-optimized kernel (where our results were obtained with) and 0.83 with CUDA-optimized kernel, where the computed time for enrollment speaker embeddings are not included. When reducing the padded context length to 25 and the number of speaker embedding vectors to 1, the RTF reduces to 0.77. Using the same hardware, our best baseline XTTS-v2 has an RTF of 0.43. Note that we have not applied techniques to speed up inference such as torch.compile or 16 or 8-bit inference. Furthermore, Mamba 2 architecture has claimed significant inference time improvement.
> Other baselines RTF: VALL-E: 0.87, MetaVoice: 2.33, YourTTS: 0.06. All values are measured using an RTX 6000 Ada Generation GPU.
>
> As long as RTF is less than 1, these differences do not have major effect on the latency, since streaming models should send out audio frames (in our case 75fps) as soon as they are generated. Our model reduces latency in a way that existing model cannot do: (1) we do not wait for a complete sentence to be formed, but start using short text chunks as they become available. (2) we generate audio to keep pace with the text stream head and avoid falling behind (e.g. when text arrives fast but the model generates slow speech)
>
> Please let us know if these have addressed your concerns or if you have other concerns. Thank you!

---

> > ### Comment · Reviewer_aFLh · 2024-11-27
> > **Thanks for your response!**
> >
> > > We hope you can elaborate on experiments that would be helpful.
> >
> > I would suggest adding an ablation study to separately synthesize 'the smaller sentences split by Coqui-TTS' using the proposed model, and then compare the performance with the one-shot results presented in Table 2.
> >
> > Furthermore, I suspect there's been a bit of a misunderstanding. The primary objective of this ablation study is to substantiate the significance of the key motivation:  using the sliding window to maintain context `across short sentences` is necessary, compared to focusing solely on the context `within each individual short sentence`.

---

> > > ### Author Response · Authors · 2024-11-27
> > >
> > > Thank you for the clarification. Yes, we believe that existing models would not work without being aware of the surrounding context and simply concatenating the outputs of short text chunks (in our case, 2-4 words). We agree that we should have provided evidence. We have included one sample of XTTS v2 on 4-word chunk similar to those used in our main experiment. The model tends to output longer silence or abrupt stop at the end of each chunk and generally less natural compared to the one which sees the whole context and ours which slides a window. We hope this is enough to address your concern. Please let us know if you have further comments!

---

> > > > ### Author Response · Authors · 2024-12-02
> > > >
> > > > As the discussion deadline approaches, we would like to address any remaining concerns to the best of our ability. Please let us know if any of the original concerns are still outstanding. We greatly appreciate feedback on the paper, as it is invaluable in helping us improve it!

---

### Official Review · Reviewer_QLwQ · 2024-11-01

**Soundness:** 2
**Presentation:** 2
**Contribution:** 3
**Rating:** 5
**Confidence:** 4

**Summary:**

The authors propose a zero-shot text-to-speech (TTS) system, called L3Speech, aming for real-time speech synthesis from continuous text chunks. Specifically, they 1) adopt a Mamba-based autoregressive decoder for linear-time generation, integrating cross-attention mechanisms utilize enrollment speech and transcript; 2) employ rotary positional embeddings within the cross-attention mechanisms to handle an infinite text stream by sliding a window; 3) apply semantic guidance to mitigate the misalignment between text and speech. The models are trained on LibriLight dataset and evaluated on the test-clean set of LibriTTS. Experimental results demonstrate that the proposed system achieves performance comparable to state-of-the-art TTS models while providing flexibility to balance latency and content accuracy in streaming scenarios.

**Strengths:**

- The proposed system accesses past and future text contexts in chunk-wise decoding, which improves the performance of steaming TTS. By incoperating semantic tokens (i.e. graphemes) alongside acoustic tokens, the authors address the problem of misalignment in autoregressive generation. These are validated by the ablation studies.
- The writing style is good, and the English is clear and concise. The provided discussions and analysis are insightful.

**Weaknesses:**

- The submitted maniscript does not include the Section of References, which hampers the reading and reviewing process.
- While the authors emphasize the advantage of linear-time decoding with the Mamba-based decoder, they don't provide ablation experiments that compare against a counterpart with transformer-based decoder. I would suggest conducting comparisons in terms of inference time or quality-speed trade-offs at different sequence lengths.
- The authors propose to use rotary positional embeddings in the cross-attention modules where the positional indices are based on arrial time, but don't provide ablation experiments about this. I would suggest comparing to using fixed positional encodings and no positional encodings.

**Questions:**

- For line 034-035, the citations are not included in parentheses.
- For line 190-191, "The speech encoder is a transformer-based encoder that converts enrollment speech of arbitrary length into a fixed-length sequence of embeddings." Could the authors provide more details on how to obtain a fixed-length sequence of embeddings?
- For line 241-242, should $\tau_{i+1}$ be $\tau_{i}+1$?
- For line 245-246, I can't understand the formulas of $p(t)$ and $f(t)$. Are they correct?
- For line 268-270, as the author mentioned "we replace each blank token with the first non-blank tokens to the right of the sequence", but in the given example, why the last two tokens in the processed sequence are still blank tokens? Shouldn't they be "c"?
- For line 270-272, could the authors provide more details about the upsampling process (from 50 tokens per second to 75 acoustic tokens per second)?
- In Algorithm 1, for line 310-311, should $p_t^{(g)}$ be the modified (scaled) one? Why is $T_{top-k}$ is not used? Steps in line 306-312 seem not fully consistent with descriptions in line 279-281.
- For Equaltion 1, the attention formula is not correct. I think the value tensors should be placed out of the softmax function.

---

> ### Author Response · Authors · 2024-11-26
> **Thank you for your response!**
>
> Thank you for your valuable feedback. We want to address your comments and questions as below:
>
> > While the authors emphasize the advantage of linear-time decoding with the Mamba-based decoder, they don't provide ablation experiments that compare against a counterpart with transformer-based decoder. I would suggest conducting comparisons in terms of inference time or quality-speed trade-offs at different sequence lengths.
>
> We understand that this is a valid concern. While we include many transformer baselines, conducting an abalation study on the backbones is challenging since they operate differently. We refer to the LiveSpeech baseline, which aims for streaming TTS and is close in terms of datasets, architectural components, the modeling of 16 Encodec codes, and the fully autoregressive nature. The benefit of Mamba in our motivation is only about inference speed (or RTF, not the latency), where a faster model helps to bring down RTF to less than 1 which is important for the model to work on device. Our RTF for the Mamba architecture on an  RTX 6000 Ada Generation is 0.83 without much inference time optimization (for reference, LiveSpeech has a RTF of 0.96). In an ideal setting, Mamba [1] claims 5x faster inference speed, and Mamba 2 [2] claims 2-8x being faster than Mamba 1. Since both backbones require complex optimization for the best inference time, we rely on the complexity and existing benchmark to justify our choice. Note that Mamba has not been dmonstrated at this large scale on the zero-shot TTS benchmark before; therefore, we believe that our results could inspire future research on choosing the better architecture.
>
> > The authors propose to use rotary positional embeddings in the cross-attention modules where the positional indices are based on arrial time, but don't provide ablation experiments about this. I would suggest comparing to using fixed positional encodings and no positional encodings.
>
> We understand the valid concern. We do not provide ablation study for the positional indices since they are not for performance, but required for our method to work. Without positional indices, our model cannot perform streaming. Positional indices based on arrival time help the model the generate audio that keeps pace with the incoming text stream. It provides an additional condition of how long a chunk should last. For example, in a text-based voice conversion task (e.g. accent conversion), if the model is not aware of the duration for each chunk, it may generate slow speech, so new text chunk keep stacking up in the queue while the model still generates for some chunk in the past. If the model generate fast speech, it finishes too soon and produces silence when waiting for the next chunk. Using the arrival time as some sort of duration condition helps our model to plan ahead and produce natural speech.
>
> We also visualize the attention map in Figure 3 and Figure 4 in the Appendix. In the first layer, we see high attention scores on chunk boundaries (the arrival time of each chunk) forming multiple short segments. In later layers, we see attention scores form a continuous line, suggesting that the audio chunk "stretches out" to keep pace with the arrival of the next text chunk to produce natural, non-interrupted speech, at the same pace with the text stream.
>
> **Questions**
>
> We appreciate your time for verifying the details in the paper and will incorporate them in the final manuscript.
>
> > For line 190-191
>
> We do this by appending $N_s$ zero vectors to the inputs and use the outputs of these positions as the encoder outputs. They are differentiated purely by fixed positional embeddings.
>
> > For line 241-242
>
> Yes!
>
> > For line 245-246
>
> Thanks for spotting this out. It should be $p(t)=\max(0, c(t)-n_p)$ and $f(t)=\min(|S|, c(t) + n_f)$. They are essentially $n_p$ chunks before and $n_f$ chunks after without going beyond the first and the last chunk.
>
> > For line 268-270
>
> We still keep blank tokens at the end in our implementation. Since for these last blank token, there is no "first non-blank tokens to the right"
>
> > For line 270-272
>
> We did something similar to interpolating a sequence of one-hot vectors and taking the argmax
>
> > In Algorithm 1, for line 310-311
>
> Sorry for the confusion. We need to add one line that $p_t(g)[k]:=0$ for $k\not\in T_{top-k}\cup T_{guiding}$. We only sample these sets.
>
> > For Equaltion 1, the attention formula is not correct. I think the value tensors should be placed out of the softmax function.
>
> Thank you for spoting out this. The value tensors are not a part of the formula, we will remove that.
>
> Please let us know if these have addressed your concerns or if you have other concerns. We will provide the updated the manuscript at the end of the rebuttal period.

---

> > ### Author Response · Authors · 2024-12-02
> >
> > As the discussion deadline approaches, we would like to address any remaining concerns to the best of our ability. Please let us know if any of the original concerns are still outstanding. We greatly appreciate feedback on the paper, as it is invaluable in helping us improve it!

---

### Official Review · Reviewer_TdUd · 2024-11-02

**Soundness:** 3
**Presentation:** 3
**Contribution:** 3
**Rating:** 6
**Confidence:** 3

**Summary:**

In this work, the authors proposed a zero-shot TTS system L3System to supports 1) infinitely long speech generation 2) text-audio stream synchronization 3) seamless transitions among the generated audio chunks. The system includes a speech encoder to generate fix-length embedding from enrollment speech audio, a text module to convert the input text into corresponding embeddings, a mamba based decoder is used to generate audio conditioned on the speech and text representations from the speech encoder and text embeddings. Only text in the related chunks are used for condition during speech generation. During training and inference, semantic guidance generated by a CTC based ASR model is used to alleviated misalignment. The model shows good results as shown in experimental sections. The audio samples provided also show good quality.

**Strengths:**

- A streaming TTS approach is provided with good results
- Semantic guidance is used to alleviate speech text misalignment.

**Weaknesses:**

- Related work is not complete. There are many streaming based TTS models have been proposed, such as Transducer based TTS [1,2], which is time synchronized based and naturally fit to the streaming applications.
- Reference section is missing
- Motivation of some method choices are not discussed, for example,  positional indices based on arrival time.
- Missing the study of the importance of acoustic model choice during soft guidance. It could be critical for languages with low resource and inferior ASR model.
- The semantic guidance is one of the contributions in this paper. It would be useful to compare the complexity during training and inference time when it is used.
- There are some typos in the paper submitted.

[1] Kim, Minchan et al. “Transduce and Speak: Neural Transducer for Text-To-Speech with Semantic Token Prediction.” ASRU (2023)
[2] Du, Chenpeng et al. “VALL-T: Decoder-Only Generative Transducer for Robust and Decoding-Controllable Text-to-Speech.” ArXiv abs/2401.14321 (2024)

**Questions:**

- How do we know the generation for one chunk is done? Is it based on the ASR model?
- Motivation to have positional indices based on arrival time? We can get that information during training when the target audio is given, how about inference, especially for $\tau_{f(t)}$?

---

> ### Author Response · Authors · 2024-11-26
> **Thank you for your response!**
>
> Thank you for your valuable feedback. We want to address your comments and questions as below:
>
> > "Related work is not complete. There are many streaming based TTS models have been proposed, such as Transducer based TTS [1,2], which is time synchronized based and naturally fit to the streaming applications."
>
> We thank you for your suggestion and will include more in related works.
>
> We highlight that we take a unique approach in reducing the latency. Most streaming models, including those suggested, require the full sentence to be available before start generating. We, however, start generating with a few word lookahead. The text transcript is usually fixed in existing models, while in our model, we keep updating the text transcript with incoming text chunks and removing old text chunks, all during a non-stop generation, without overhead of changing the context.
>
> Take a text-based accent conversion as an example, existing models may need to wait for a sentence to be finsihed to start generating, and may struggle to maintain smooth transitions between turns if turns are generated independently. Moreover, there is no way to control the duration of a sentence (if the TTS produces slow speech, it won't be able to keep pace with more incoming text chunks). Our approach start using whatever an ASR model outputs and make sure that it keeps pace with the most recent text chunk.
>
> > Motivation of some method choices are not discussed, for example, positional indices based on arrival time.
>
> As discussed above, our model needs to keep pace with the most recent text chunk in a non-stop text stream. The arrival time is an input from the upstream task indicating how long we should generate speech for a specific chunk. When there is no text transformation, a chunk that lasts 3s in the input audio stream should last 3s in the output audio stream. When there is some text transformation (e.g. translation), we also expect the same constraint or the output stream may quickly become out of sync with the input stream.
>
> > Missing the study of the importance of acoustic model choice during soft guidance. It could be critical for languages with low resource and inferior ASR model.
>
> We only target English in the paper, which can be extended to other popular languages. Our soft guidance may need special adaptation for some languages; however, if the language can be phonemized, we can use that as the unit. ASR models such as wav2vec2-phoneme is trained on multi lingual corpus and can generalize phonemes across different languages. During training, our model generates aligned phonemes together with audio frames. During inference, we need to convert any text transcript to phonemes and use that to guide speech generation.
>
> **Questions**
>
> > How do we know the generation for one chunk is done? Is it based on the ASR model?
>
> If a text chunk needs time $t$ to retrieve (counting from the arrival of the previous chunk to the arrival of the current chunk), its generated speech will last for time $t$.
>
> > Motivation to have positional indices based on arrival time? We can get that information during training when the target audio is given, how about inference, especially for $\tau_{f(t)}$?
>
> We rely on the upstream task to provide the duration for each text chunk. During training, we randomly split long text into chunks based on the word boundaries obtained with CTC alignments, where the boundaries can be a random time frame in between two words.
>
> If our upstream task requires little to no text transformation (voice/accent conversion, disfluency removal, etc.), the ground-truth boundaries are available during inference. If some text transformation is involved (e.g. speech translation), we still expect our model to be robust to small adjustments in the boundaries. We provide additional experiment results when we add $t_r$, a maximum random time delta in seconds for each chunk boundary.
>
> | $t_r$ | CER | SS | O-MOS |
> | --| ------ | ---- | ----------- |
>  0.0 | 2.7 | 70.5 | 3.99
> 0.2 | 3.6 | 69.5 | 3.95
> 0.4 | 5.3 | 68.5 | 4.00
>
> Note that a chunk has only 2-4 words. We see some degradation within a reasonable range when the provided duration may be too short or too long for a certain text chunk. However, this is also a difficult scenario that we can benefit from more finetuning.
>
> We believe that this audio-text stream synchronization is necessary for a production streaming system. An example is that "thank you" in English is translated to "arigatou gozaimasu" in Japanese which tends to produce a longer audio chunk. Our system makes sure that the Japanese phrase is spoken as fast as "Thank you" so that future chunks are not delayed by the playback of the current chunk.
>
> Please let us know if these have addressed your concerns or if you have other concerns. We will provide the updated the manuscript at the end of the rebuttal period.

---

> > ### Comment · Reviewer_TdUd · 2024-11-26
> > **Thanks for the replies**
> >
> > > Most streaming models, including those suggested, require the full sentence to be available before start generating.
> >
> > It is not true for the Transducer based TTS systems aforementioned.
> >
> > >If a text chunk needs time  $t$ to retrieve (counting from the arrival of the previous chunk to the arrival of the current chunk), its generated speech will last for time $t$.
> >
> > What will happen if $t$ is too long?

---

> ### Author Response · Authors · 2024-11-27
>
> Thank you for your responses! We address these two points as below:
>
> > It is not true for the Transducer based TTS systems aforementioned.
>
> Upon reviewing the transducer architecture in zero-shot TTS more carefully, we agree with you that these models do not need the full sentence and are relevant for Related Works. However, these works focus on the monotonic alignment of the transducer and are still not streaming due to the non-autoregressive inference of fine codebooks (e.g. 2nd to 8th codebooks), and the lack of duration control. Moreover, scaling the transducer can be challenging, as evidenced by the fact that these studies report results only on LibriTTS. Our hard/soft guidance similarly promotes monotonic alignment while still benefitting from the transformer architecture (e.g., scaling). We believe that future work could benefit from fully adapting the architecture for streaming and demonstrating its effectiveness in large-scale zero-shot scenarios. We thank you for pointing this out and will discuss these works in the Related Works sections
>
> > What will happen if $t$ is too long?
>
> Our model may not work for an unrealistic value of $t$. Usually, $t$ is from an input audio stream and an upstream model is responsible for streaming text chunks compatible with our the supported range. If $t$ is too long, does it mean the input audio contains mostly silence? In such case, the upstream task should break silence to multiple empty chunks.
>
> We believe that some sort of duration control is neccessary for the task. For example, in a voice conversion task, if the enrollment speech is slow but the source speech is fast, the output speech is likely to be slow similar to the enrollment speech and the accumulated latency can become large as the speech stream progresses. We believe our approach to control the speech duration based on the text arrival time is the most straightforward way to handle this; a fixed or no duration control would not work in a non-stop streaming scenario.

---

> > ### Author Response · Authors · 2024-11-27
> >
> > We also missed one of your points:
> >
> > > The semantic guidance is one of the contributions in this paper. It would be useful to compare the complexity during training and inference time when it is used.
> >
> > During training, we only increase the number of embedding layers and projection layers from 16 to 17 to accommodate grapheme/phoneme tokens. This adds around 50k parameters to the model and should not slow down training.
> >
> > During inference, semantic guidance runs entirely on CPU with CER being the major computation. We believe that it should be very fast and incur no latency since it can also be run in parallel with model inference (while the model performs a forward pass, semantic guidance refines the guiding candidates)
> >
> > Please let us know if these have addressed your concerns or if you have other concerns. We sincerely appreciate the time you have dedicated to reviewing and engaging in the rebuttal process.

---

> > > ### Author Response · Authors · 2024-12-02
> > >
> > > As the discussion deadline approaches, we would like to address any remaining concerns to the best of our ability. Please let us know if any of the original concerns are still outstanding. We greatly appreciate feedback on the paper, as it is invaluable in helping us improve it!

---

### Official Review · Reviewer_yudk · 2024-11-03

**Soundness:** 3
**Presentation:** 4
**Contribution:** 3
**Rating:** 6
**Confidence:** 4

**Summary:**

This paper addresses the challenge of zero-shot text-to-speech (TTS) in continuous streaming scenarios, where text input is provided in short chunks, necessitating real-time, low-latency responses. The authors propose L3Speech, a stream-aware model incorporating several novel components: a Mamba-based decoder for efficient, linear-time decoding; rotary positional embeddings to manage continuous text flow; and semantic guidance using grapheme tokens to maintain alignment between text and speech. Experimental results show that L3Speech provides flexibility in streaming scenarios, achieving competitive or superior performance to existing TTS models in terms of content accuracy, speaker similarity, and overall audio quality.

**Strengths:**

- The paper is well-written and easy to follow.

- This paper addresses an emerging and impactful area—streaming TTS with integration into large language models. By targeting the latency challenges and supporting flexible, continuous input, the model shows promise for real-time applications.

- The proposed architecture is well-grounded, utilizing the Mamba-based decoder, rotary positional embeddings, and semantic guidance via grapheme tokens. These design choices are technically solid, and their contributions to the model's performance are validated through an extensive ablation study.

- The model supports a range of lookback and lookahead settings, making it adaptable to various streaming TTS applications. This flexibility is an essential feature for practical deployment across different tasks and latency requirements.

**Weaknesses:**

- Missing reference section.
- Lack of Real-Time Runtime Analysis: A runtime analysis comparing the proposed method with baselines—especially in scenarios where L3Speech is cascaded with a language model—would provide a clearer picture of the latency benefits. Including this analysis would strengthen the paper by illustrating the real-world efficiency gains.

**Questions:**

Baseline Comparison with Transformer Decoder: Incorporating a baseline using a transformer-based decoder could further validate the performance and efficiency advantages of the Mamba architecture. There are well-known techniques to improve the inference speed and local attention mechanism.
This addition would provide a stronger basis for the choice of Mamba over more conventional architectures.

---

> ### Author Response · Authors · 2024-11-26
> **Thank you for your response!**
>
> Thank you for your valuable feedback. We want to address your comments and questions as below:
>
> > Lack of Real-Time Runtime Analysis
>
> Measuring the end-to-end latency is challenging in our case, since it heavily depends on upstream tasks. Our goal is to propose a flexible approach that can be adapted to any streaming conditions with the best possible latency. We provide some scenarios with latency analysis as below:
>
> - Voice Conversion: we consider text-based voice conversion (which can also be challenging tasks such as accent conversion, whisper to speech conversion) as the upstream task, where the model outputs word by word. With chunk length of 1 and lookahead of 1 (extreme cases need furthur finetuning as reported in Table 13&14), our latency is 2 words
> - Live Translation: we consider an upstream model that can output text chunks instead of full sentences (e.g. SeamlessM4T). If each chunk has an average of 4 words and there is one chunk lookahead, our latency is 8 words. Our approach also benefits from speech generated to keep pace with the text stream head, which is non measurable. For example, starting from the time 0, if the first chunk arrives at the time $t_1$, but needs $t'_1>t_1$ to play, the playback of the second chunk will be delayed by the audio-text duration gap
> $|t'_1-t_1|$. We remove this latency by making sure that $t'_1=t_1$ (audio-text stream synchronization)
> - Voice Assistant: adding voice to an LLM with our model is trickier since if LLM generates text fast enough, there will be sufficient lookaheads most of the time. We consider a scenario when LLM generates words at the same pace with the speech (e.g., large model running on device), in which our model aims to follow the latest words from LLM. Similar to voice conversion, our model can have as low as 2 word lookahead.
>
> In these examples, we ignore the time-to-first-frame, which should be as fast as a single step, since the model can stream out audio frames as soon as they are generated, and adding text chunks introduces zero overhead (only embedded, not encoded).
>
> > Baseline Comparison with Transformer Decoder: Incorporating a baseline using a transformer-based decoder could further validate the performance and efficiency advantages of the Mamba architecture. There are well-known techniques to improve the inference speed and local attention mechanism. This addition would provide a stronger basis for the choice of Mamba over more conventional architectures.
>
> We rely on the original papers [1] and [2] to justify our choice for Mamba in terms of inference speed and computational complexity, which is critical to address in on-device streaming settings. Our hypothesis is that a full access to long context length may not be required for the TTS task, since the content and the speaker characteristic are accessible via cross attention and the model only needs local context to ensure continuous generation. With that hypothesis, a transformer with local attention may work as well as Mamba in terms of speed and performance. We only demonstrate the performance in our experiments, where the main observation is that Mamba can produce results as good as competitive transformer baselines.
>
> [1] Mamba: Linear-Time Sequence Modeling with Selective State Spaces [2] Transformers are SSMs: Generalized Models and Efficient Algorithms Through Structured State Space Duality
>
> Please let us know if these have addressed your concerns or if you have other concerns. We will provide the updated manuscript at the end of the rebuttal period.

---

> > ### Author Response · Authors · 2024-12-02
> >
> > As the discussion deadline approaches, we would like to address any remaining concerns to the best of our ability. Please let us know if any of the original concerns are still outstanding. We greatly appreciate feedback on the paper, as it is invaluable in helping us improve it!

---

### Official Review · Reviewer_AQZF · 2024-11-04

**Soundness:** 1
**Presentation:** 3
**Contribution:** 2
**Rating:** 3
**Confidence:** 3

**Summary:**

The paper presents L3Speech, a stream-aware zero-shot text-to-speech (TTS) model designed for continuous text input in short chunks, which is crucial for streaming applications. It leverages Mamba for linear-time decoding, rotary positional embeddings for infinite text stream processing, and semantic guidance for efficient text-audio synchronization. Experimental results demonstrate L3Speech's competitiveness with some other TTS systems while offering flexibility for various streaming scenarios.

**Strengths:**

The paper introduces L3Speech, a Mamba-based TTS system designed for continuous text input in short chunks, essential for streaming applications.

1. This paper explores the use of Mamba as the backbone for AR-based TTS.

2. It proposes a decoding method assisted by semantic guidance to enhance model stability.

3. Various tricks, such as improved rotary positional embeddings, are designed to improve the model's performance in streaming inference, including streaming text input.

**Weaknesses:**

The paper introduces L3Speech, a Mamba-based TTS system designed for continuous text input in short chunks, essential for streaming applications. It proposes techniques such as rotary positional embeddings for infinite text stream processing, semantic guidance, and optimized text chunk length and quantity selection. However, it has notable weaknesses, including:

1. Although this paper claims to support infinite text input and streaming inference, it only evaluates on the LibriTTS dataset. I think this dataset cannot truly demonstrate the model's benefits for long text input and streaming inference. The authors should evaluate on more suitable test sets. Additionally, the baselines chosen for comparison are relatively weak, as they do not include more advanced models like VoiceBox, NaturalSpeech 3, and CosyVoice.

2. The authors did not demonstrate the necessity of the Mamba architecture, I think transformers could also achieve the same methods.

**Questions:**

1. Do you evaluate the model on alternative datasets that would be more suitable for evaluating the performance of L3Speech in streaming scenarios?

2. Do you compare the model with more powerful TTS systems?

3. What evidence or experiments could you provide to demonstrate the necessity of the Mamba architecture over conventional transformers for achieving the proposed methods?

4. How might the performance of L3Speech differ if implemented with a conventional transformer architecture instead of Mamba?

---

> ### Author Response · Authors · 2024-11-16
> **Thank you for your valuable feedback!**
>
> Thank you very much for your valuable feedback. We sincerely appreciate the constructive feedback provided. However, we believe that the inclusion of additional test sets and baselines is not necessary for addressing the points presented in the paper. We address weakness points as below:
>
> > “I think this dataset cannot truly demonstrate the model's benefits for long text input and streaming inference. The authors should evaluate on more suitable test sets”
>
> In our experiments, we demonstrated that our model can operate *beyond the maximum context length that it was trained on*. Our experiments use a filtered set of 10-20s samples (avg: 14.7s), while our model is trained on maximum 10s. We believe that effective shows that our model can handle infinitely long text stream by sliding a window, since our model *does not have access to the absolute position of text tokens* (in the same way as RoPE or a CNN/RNN network). We will, however, include a few longer samples in the attachment for qualitative listening.
>
> > “Additionally, the baselines chosen for comparison are relatively weak, as they do not include more advanced models like VoiceBox, NaturalSpeech 3, and CosyVoice.”
>
> Our strongest baseline is XTTS v2, which is currently the 4th model in the TTS Arena (https://huggingface.co/spaces/TTS-AGI/TTS-Arena), from which there are industrial candidates. We compare our model to only those that we can have access to a checkpoint since many factors can affect the results (e.g. enrollment speech can be chosen differently, not every paper uses the entire LibriTTS test sets, etc.)
>
> Furthermore, we only compare our model to language model-based approaches. Among suggested candidates, VoiceBox and Natural Speech 3 are diffusion-based and may not be suitable for streaming since the whole utterance is generated non-autoregressively, while more recent work like CosyVoice is not required for comparison under the ICLR guidelines (within 4 months). We believe that some of our baseline models (MetaVoice, XTTS v2) are among the best open-source candidates and others (SpeechX, LiveSpeech) are models trained with similar training data and model size. Our primary focus is also to demonstrate the streaming capabilities of our model, rather than achieving state-of-the-art performance.
>
> > “The authors did not demonstrate the necessity of the Mamba architecture, I think transformers could also achieve the same methods.”
>
> Mamba is not required functionally; however, this aims to reduce the inference time and allow the model to run on device (where streaming models usually run on), which is critical for streaming since the model needs to produce audio tokens at 75Hz. We identify the inference speed (different to latency) as one of challenges for on-device streaming (Introduction / paragraph 2 / last sentence) and propose Mamba to alleviate that. We refer to the original work [1,2] for an extensive comparison between the Mamba and the transformer architecture in terms of inference speed (as for your requested evidence). More importantly, we have showed that Mamba performs comparably to transformer counterparts overall. We want to highlight the fact that the Mamba architecture has not been used for large scale (>60k hours) training of zero-shot TTS models before, so validating the feasibility of the model is an important contribution for future streaming TTS research.
>
> [1] Mamba: Linear-Time Sequence Modeling with Selective State Spaces
> [2] Transformers are SSMs: Generalized Models and Efficient Algorithms Through Structured State Space Duality
>
> We hope these have also answered your questions. By responding early, we look forward to hearing from you to confirm if these points have been addressed satisfactorily!

---

> > ### Comment · Reviewer_AQZF · 2024-11-24
> >
> > Can you provide a link for some longer samples?

---

> > > ### Author Response · Authors · 2024-11-26
> > >
> > > Thank you for your request! We will update the manuscript and include these longer samples (>1min) in the attached supplementary materials. We plan to do that by Nov 27 after we incorporate all experiments and comments to avoid updating multiple times.
> > >
> > > If you have additional concerns, please let us know so we will try to address them timely. Thank you!

---

> > > > ### Author Response · Authors · 2024-12-02
> > > >
> > > > As the discussion deadline approaches, we would like to address any remaining concerns to the best of our ability. Please let us know if any of the original concerns are still outstanding. We greatly appreciate feedback on the paper, as it is invaluable in helping us improve it!

---

### Author Response · Authors · 2024-11-13
**References updated**

Dear Reviewers,

We apologize for the omission of the References section in the previous version. We have now updated the preprint to include this section, with the main content unchanged.

Thank you for your valuable feedback. We will be providing our detailed responses shortly.

Best regards,

---

### Author Response · Authors · 2024-11-27
**Manuscript & supplementary material updated**

Dear Reviewers,

We have updated our manuscript incorporating reviewers' comments and include some audio files in the attached supplemental material. The new audio files include:
- 5 samples (/audio/longer) of longer than 1 minute duration, these can be used to test if there is any degradation over time
- one sample (/audio/chunk_by_chunk) demonstrates the results of running XTTS-v2 inference chunk-by-chunk, highlighting that generating chunks independently with a non-streaming model produces unsatisfactory outcomes. This underscores the necessity of our sliding window approach to ensure smooth and natural transitions between chunks.

Additionally, we hope that we have addressed the majority of the reviewers' concerns in each review. We welcome any further discussion and additional concerns from the reviewers. Regardless of the final scores, we sincerely appreciate the time and effort the reviewers have dedicated to reviewing and engaging in the discussion.

---

### Meta-Review · Area_Chair_6RV9 · 2024-12-17

**Metareview:**

The paper introduces L3Speech, a zero-shot text-to-speech (TTS) system tailored for real-time streaming scenarios. It supports continuous text input in short chunks, enabling infinite speech generation with seamless transitions between audio segments. Key innovations of the system include: 1) Mamba-based Decoder for linear-time autoregressive decoding, ensuring efficient real-time audio generation. 2) Rotary Positional Embedding integrated within cross-attention mechanisms to handle infinite text streams effectively by sliding a context window. 3) semantic Guidance employed during training and inference using a CTC-based ASR model to address text-audio misalignment and improve synchronization.

The system is trained on the LibriLight dataset and evaluated on the test-clean set of LibriTTS. Experimental results demonstrate that L3Speech achieves performance comparable to or surpassing SOTA TTS systems in terms of content accuracy, speaker similarity, and audio quality. It also offers flexibility to balance latency and performance, making it well-suited for various streaming scenarios.


Strength of this paper
The paper explores a novel and impactful area by addressing streaming text-to-speech (TTS) challenges, particularly in zero-shot settings, making significant contributions to real-time applications. Key strengths include:
- Innovative Architecture: utilizes a Mamba-based decoder for efficient linear-time autoregressive (AR) decoding, validated as feasible for zero-shot TTS tasks; incorporates rotary positional embeddings tailored for streaming inference, enhancing performance with continuous text input; employs semantic guidance, leveraging grapheme tokens to alleviate text-speech misalignment and improve stability.
- Practical and Flexible Design: Proposed method supports chunk-wise decoding with adjustable lookback and lookahead settings, making it adaptable to various latency and task requirements in streaming TTS applications.
- Ablation studies show efficacy of proposed method, which demonstrate competitive performance comparable to SOTA TTS systems.

Weakness of this paper
- Incomplete Comparisons and Analyses: 1) Ablation Studies: Key components, such as the necessity of the Mamba-based decoder, rotary positional embeddings (based on arrival time), and semantic guidance, lack sufficient ablation studies against alternatives like transformer-based decoders, fixed positional encodings, or other baseline architectures. 2) Latency and Runtime Analysis: Missing comparisons of latency benefits, particularly in real-world scenarios where the system interacts with language models. A detailed runtime analysis would strengthen the argument for the proposed approach. 3) Some design choices, such as positional indices based on arrival time and the importance of the acoustic model in semantic guidance, lack clear motivations or validation. The claim of "infinitely long speech streaming" has not been thoroughly demonstrated through experiments involving long text streams or comparisons between full and incomplete text streams.
- Experimental and Reproducibility Issues: The evaluation is restricted to the LibriTTS dataset, which may not adequately demonstrate the model's advantages for long text input and streaming inference. Testing on more suitable datasets and advanced baselines (e.g., VoiceBox, NaturalSpeech, CosyVoice) is needed. Besides, key experimental details are missing, affecting the reproducibility of the system. Additional experiments, such as testing robustness on longer text streams or comparing seamless chunk transitions, are needed to validate claims.
- Lastly, related work and references is not incomplete, e.g., overlooking streaming-based TTS models like transducer-based TTS approaches. This makes it harder to understand the position of this work, and what's the trade-offs and how proposed work compared with existing approaches, to help readers better understand the method's limitations and future applicability.

**Additional Comments On Reviewer Discussion:**

Although some of these weakness have been improved / somewhat addressed during rebuttal session, I do feel the session is too short and I would like to see a more comprehensive modification to systematically working on these suggestions. The rating is still low after rebuttal session. Thus, I recommend to reject the work, and the authors to re-work on these weakness and re-submitting to future conferences.

---

### Decision · Program_Chairs · 2025-01-22

Reject